# In situ enzymatic control of colloidal phoresis and catalysis through hydrolysis of ATP

Ekta Shandilya[1], Bhargav Rallabandi [2]✉ & Subhabrata Maiti [1]✉

The ability to sense chemical gradients and respond with directional motility and chemical activity is a defining feature of complex living systems. There is a strong interest among scientists to design synthetic systems that emulate these properties. Here, we realize and control such behaviors in a synthetic system by tailoring multivalent interactions of adenosine nucleotides with catalytic microbeads. We first show that multivalent interactions of the bead with gradients of adenosine mono-, di- and trinucleotides (AM/D/TP) control both the phoretic motion and a proton-transfer catalytic reaction, and find that both effects are diminished greatly with increasing valence of phosphates. We exploit this behavior by using enzymatic hydrolysis of ATP to AMP, which downregulates multivalent interactivity in situ. This produces a sudden increase in transport of the catalytic microbeads (a phoretic jump), which is accompanied by increased catalytic activity. Finally, we show how this enzymatic activity can be systematically tuned, leading to simultaneous in situ spatial and temporal control of the location of the microbeads, as well as the products of the reaction that they catalyze. These findings open up new avenues for utilizing multivalent interaction-mediated programming of complex chemo-mechanical behaviors into active systems.

Biological systems exploit external stimuli to regulate feedback loops and modulate biocatalytic reactions within cells in both time and space[1,2]. To create synthetic systems that are similarly sophisticated to biological ones, it is necessary to exert precise spatial and temporal control of catalytic processes, and to endow them with complex properties such as adaptability, responsiveness to stimuli, and dynamic specificity towards specific receptors or reactions[3]. In such systems, a catalyst must be autonomously relocated and toggled between on and off states in situ. This is a formidable challenge, surmounting which is crucial for the development of spatially segregated chemistry to synthesize novel products or complex structures that are not accessible using conventional methods. Recent work has sought to understand and control the transport of colloidal objects—ranging from nanometer-sized enzymes to the micron-sized droplets and polymeric beads—in a variety of chemical gradients[4–11] Here, we show how

multivalent interactions of catalytic colloids with adenosine nucleotides simultaneously govern both transport and chemical reactivity. We then demonstrate how these interactions can be controlled in situ with enzymes, to achieve precise spatiotemporal control of catalytic activity.

Multivalent interactions are fundamental to both biological and synthetic supramolecular processes[12–18]. This property has been used for sensing and catalysis, the design of drug molecules, altering biochemical pathways to modify viral and cellular behaviors, and to generate dynamic self-assembled systems[19–32]. On the other hand, multivalent interactions in ion gradients have been used to drive the phoretic motion of charged microparticles[33,34]. The physicochemical origin of diffusiophoretic colloidal transport in gradients of salts was established by Derjaguin, and Anderson et al. [35–44]. This transport has been studied extensively in systems involving common inorganic salts[39–50], and more recently in gradients of pH[38]. In cellular systems, the

[1]Department of Chemical Sciences, Indian Institute of Science Education and Research (IISER), Mohali, Knowledge City, Manauli 140306, India. [2]Department of Mechanical Engineering, University of California, Riverside, CA 92521, USA. ✉e-mail: bhargav@engr.ucr.edu; smaiti@iisermohali.ac.in

phoresis of large molecules due to gradients of small molecules such as metabolites or ATP is ubiquitous, and has gained recent attention[51,52].

We investigate multivalent interaction-mediated phoretic transport of a fluorescent, cationic micron-sized bead (abbreviated CMB) exposed to gradients of (mixtures of) nucleotides, namely adenosine mono/di/triphosphate (AMP/ADP/ATP). The CMB consists of a carboxylic acid-modified polystyrene fluorescent bead electrostatically bound with cationic cetyltrimethylammonium bromide (CTAB)-coated gold nanorods (GNR) (Fig. 1a)[53]. Additionally, the CMB enable a proton transfer, and thus catalyze the Kemp Elimination (KE) reaction in the presence of ATP and AMP (Fig. 1a)[54]. AMP activates the CMB-catalyzed KE reaction whereas ATP deactivates it. This property has been used in the past by us and others to probe enzymatic reaction mechanisms, to develop model catalysts, to control pH in a gel matrix, etc[55–60].

Here, we exploit the dual role of multivalent interactions of CMB with nucleotides, achieving simultaneous control of (i) phoretic motion, and (ii) catalytic activity. Furthermore, we show how both effects can be tailored in situ by downregulating these interactions via enzymatic hydrolysis of ATP to AMP. In particular, we developed an autonomous system that exhibits a rapid increase of phoretic velocity (which we term a phoretic jump) through enzymatic activity, and show how this effect can be utilized to program the positioning of the catalytic particles. We combine this spatial control with the nucleotide-selective catalytic activation behavior described above to develop a model system in which catalytic activity—and thus reaction products—can be regulated by colloidal phoresis both in space and time. In the remainder of this article, we develop and characterize this complex chemo-mechanical system, after first isolating the phoretic and catalytic processes individually.

## Results and discussion
### Nucleotide-binding ability of CMB
We synthesized catalytically active CMB as reported in the literature (details in supplementary information (SI))[53], by starting with a carboxylate-functionalized polystyrene particle with a hydrodynamic diameter of $1 \mu m$ and a zeta potential of $-90 \pm 5$ mV. Upon conjugation

with cationic GNR (length $23 \pm 5$ nm, width $6 \pm 1$ nm, and $\zeta = 100 \pm 10$ mV), the effective zeta potential of the fluorescent CMB reverses to +80 mV (nearly that of the GNR) indicating the binding and simultaneous reversal of the effective surface charge of the fluorescent CMB (Fig. 1b, c, Supplementary Fig. 1 and 2, SI). As expected, the zeta potential of CMB (which is a measure of surface charge) decreased in the presence of 1 mM nucleotides to $47 \pm 8$, $40 \pm 5$ and $9 \pm 4$ mV with AMP, ADP, and ATP, respectively, indicating surface binding (Fig. 1d, Supplementary Fig. 3, SI). Of all of the nucleotides, ATP binds most strongly due to the simultaneous interaction of three phosphate groups with the GNR on the CMB surface (Supplementary Fig. 4, SI)[23]. Binding with ATP does not affect the stability of the CMB as no detachment of GNR from the bead was observed (Supplementary Fig. 6, SI). The catalytic activity of the CMB is detailed in later sections.

### Diffusiophoretic drift of CMB in a gradient of nucleotides—experiments and theory
The positively charged CMB exhibit diffusiophoretic motion in gradients of adenosine nucleotides (Fig. 2). We studied this diffusiophoresis both with individual nucleotides of different valency, and with mixtures of nucleotides that compete to bind with the surface of the CMB. We set up a microfluidic experiment with a two-inlet and one-outlet channel of width $w = 600 \mu m$ and height $h = 100 \mu m$, while injecting CMB and 1 mM nucleotide solution into two inlets, each at a flow rate of $Q/2 = 150 \mu l/h$ (Fig. 2a, details in Methods). The CMB were observed to drift across the channel towards the nucleotides as they flowed through the channel, while no such drift was observed in the absence of nucleotides (Fig. 2b). We measured the concentration profiles of CMB across the channel width, at a location 1.6 mm downstream of the inlets, and then normalized these concentrations by their integral across the channel, reflecting the fact that the number of particles at any cross-section remains constant (Fig. 2c). We define the cross-stream drift distance as the location at which the normalized particle concentration profiles attain a value of 0.1 (under this definition, the maximum possible drift distance is the half-width of the

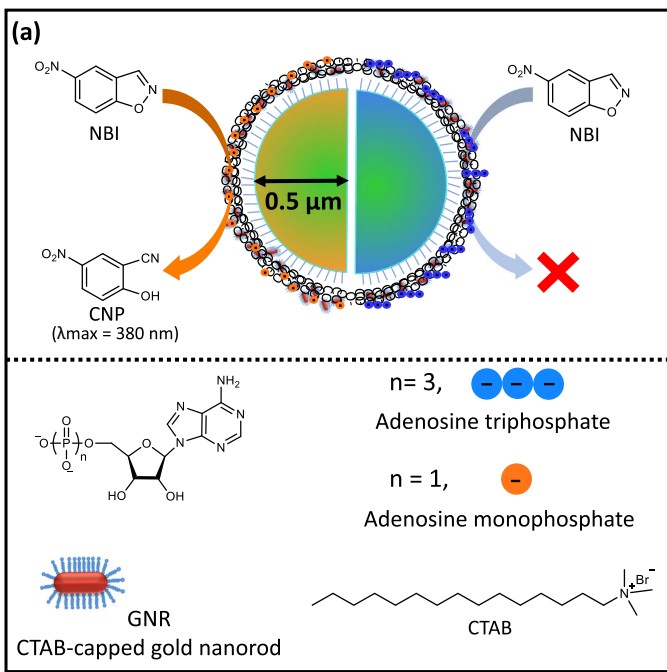

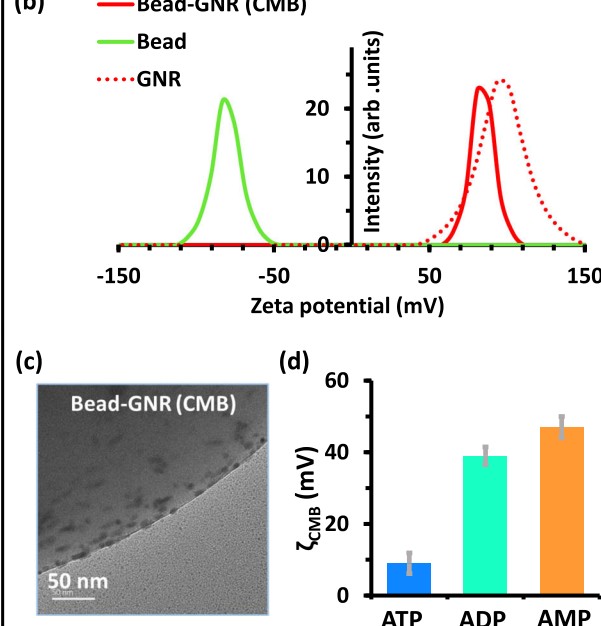

**Fig. 1 | Characterization and nucleotide binding ability of the colloid.**
**a** Schematic representation of a carboxylate-functionalized polystyrene bead modified with gold nanorods (CMB) catalyzing the Kemp elimination (KE) reaction. KE product (2-cyano nitrophenol (CNP)) is formed after addition of AMP but not ATP. **b** Zeta potential profile of CMB, only beads, and GNR in water at 25 °C. **c** TEM image of CMB conjugate showing GNR-bound bead surface. **d** Zeta potential of CMB in the presence of adenosine-based nucleotides (1 mM). Data are presented as mean ± standard deviation (SD) ($n = 5$). Source data are provided as a Source Data file.

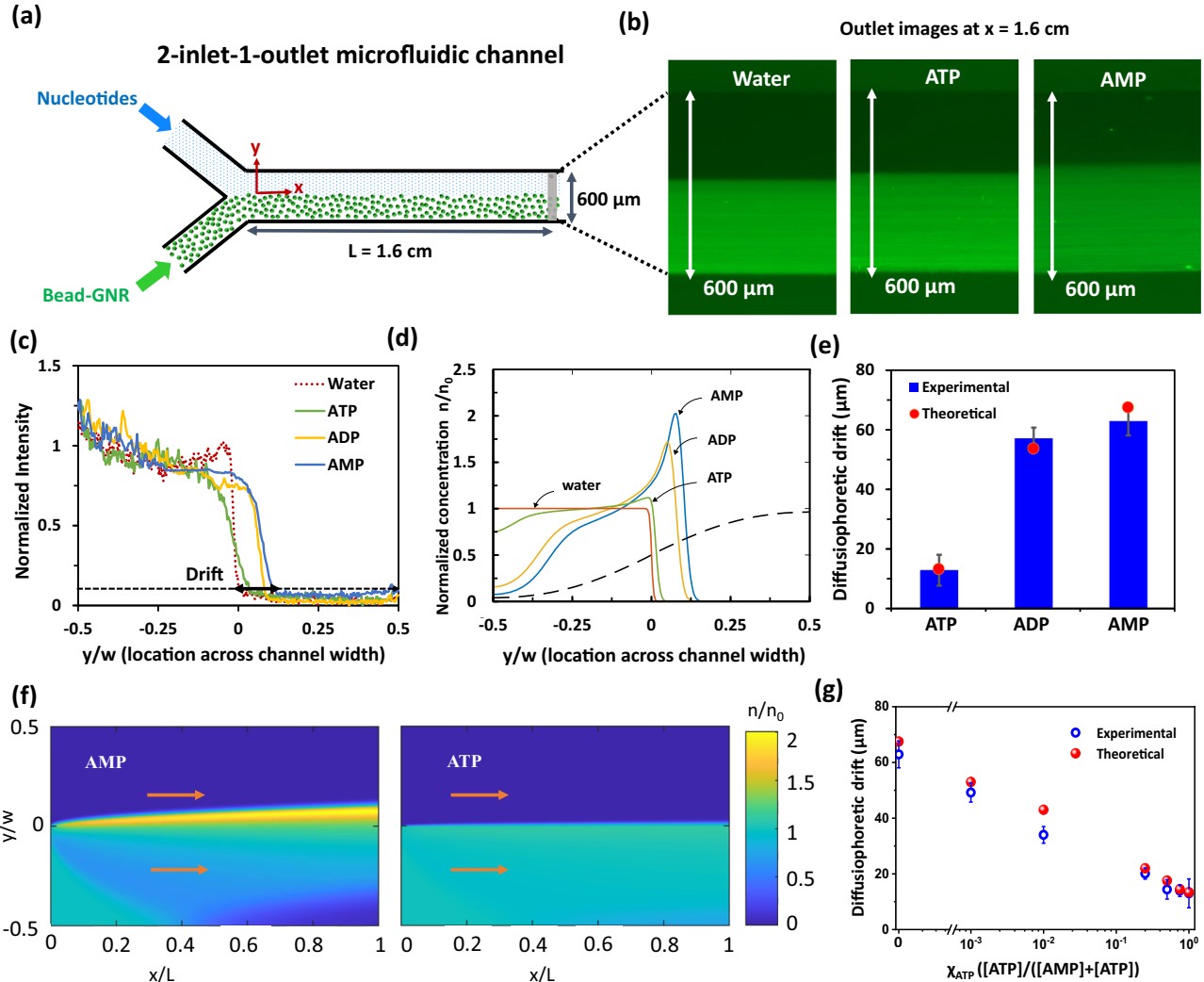

**Fig. 2 | Diffusiophoretic drift of cationic micron-sized bead (CMB) in gradient of nucleotides under catalytic and non-catalytic conditions. a** Schematic of the 2-inlet-1-outlet microfluidic setup, where CMBs flow through one inlet and nucleotides from the other. **b** Representative fluorescence image of the channel at the outlet (1.6 cm from the inlet) showing that the drift of the fluorescent CMB towards AMP is greater than ATP and control. **c, d** Normalized fluorescence intensity profile due to CMB at the outlet from the bottom to the top of the channel in (**c**) experiments and (**d**) the model (**d**). Particles drift due to diffusiophoresis towards higher concentrations of nucleotide (dashed curve in (**d**)) and the magnitude of drift follows the ordering AMP > ADP > ATP, consistent with experiments.

**e** Experimental and theoretical diffusiophoretic drift of CMB towards AMP, ADP and ATP show excellent agreement. Phoretic drift was calculated at a normalized intensity of 0.1 (near the baseline) as denoted by dotted line in Fig. 2c. **f** Theoretically calculated maps of particle concentration within the channel for AMP and ATP. The particle drift under exposure to ATP is negligible (**g**) Experimental and theoretical diffusiophoretic drift of CMB in a mixture of AMP and ATP are in good agreement across different mole fractions of ATP $\chi_{ATP}$. All experimental data are presented as mean ± SD (where $n = 4$). Source data are provided as a Source Data file.

channel, 300 μm); see Fig. 2c. We find that particles drift toward the nucleotide (up the gradient) in the cases of AMP and ADP by 63 ± 4 and 57 ± 3 μm, respectively, and by much less (only about 13 ± 5 μm) with ATP (Fig. 2e, Supplementary Figs. 8–11, supplementary Table 3). These significant drift distances (as large as 21% of the half-width of the channel for AMP) are much greater than the variability from trial to trial (see Supplementary information).

We quantitatively confirm that diffusiophoresis is the mechanism driving this particle motion by developing a two-dimensional height-averaged model of the transport of fluid, nucleotides, and particles (Fig. 2d–f). We model the transport of nucleotides by an advection-diffusion equation, which establishes a gradient of nucleotides across the channel (see Methods). The CMB are transported by a combination of fluid flow and diffusiophoretic motion (which occurs primarily across the channel). We account for both chemiphoresis (motion driven by an osmotic pressure gradient across the particle) and electrophoresis

(motion due to a local electric field generated by a gradient of cations and anions with different diffusivities) (Supplementary Table 2, SI)[36,42–47]. Diffusion of the CMB is included in the model for completeness but plays a negligible role. The competition of advection with diffusion and diffusiophoresis establishes a steady-state distribution of nucleotide and particle concentrations. The calculated steady-state concentrations near the end of the channel ($x$ = 1.6 mm from the inlet) are plotted in Fig. 2d, showing a drift of the CMBs towards the nucleotides. As with experiments, AMP produces the greatest drift, followed by ADP and ATP. We follow the experimental protocol and compute the cross-stream drift distance in the model and find excellent quantitative agreement with the experiments (Fig. 2e). We remark that the model does not involve any fitting parameters, and all model inputs (in particular, diffusivities and zeta potentials) are extracted directly from experimental measurements (see SI and methods section for details). Because the transport of CMB is controlled by diffusion of nucleotides

(diffusivity $D$), the drift distance scales with $\left(\frac{Dxwh}{Q}\right)^{1/2}$ (with prefactors involving the diffusiophoretic mobility of the particle; see Methods), up to the channel half-width $w/2$.

We repeated the experiments described above with a 1 mM *mixture* of AMP and ATP nucleotides, and studied the drift distance as a function of the fraction of ATP, $\chi_{ATP} = [ATP]/([ATP] + [AMP])$ (Supplementary Figs. 12–16, Supplementary Table 4, SI). The drift was about 63 μm in AMP ($\chi_{ATP} = 0$) as noted previously, and decreased sharply with the addition of ATP, dropping by over 50% with as little as 10 μM ATP (the mixture also contains 990 μM of AMP; $\chi_{ATP} = 0.01$), and down to 20 μm at $\chi_{ATP} = 0.25$. These findings are reproduced with excellent quantitative accuracy by the modeling framework, again with no fitting parameters (Fig. 2g). As before, we use experimentally measured zeta potentials (see supplementary Fig. 5 for these measurements) as inputs to the model, but we now track the transport of individual nucleotides. We then relate their concentration gradients to the particle motion using the well-established diffusiophoretic theory for multivalent mixtures (see Supplementary Information), again accounting for both electrophoretic and chemiphoretic contributions[33,39,49].

These observations clearly demonstrate that a small fraction of ATP greatly suppresses the AMP-mediated phoretic motion of the CMB. ATP binds much more strongly than AMP due to multivalent interactions (simultaneous interaction of multiple charged phosphates with the bead); similar selective binding of higher valent nucleotides on cationic nanoparticle surfaces, even in the presence of lower valent nucleotides, has been reported in other systems[22,54]. Therefore, during an ATP/AMP mixture experiment, the zeta potential of the surface is strongly controlled by ATP, even at relatively low mole fractions. The model confirms that this lowered zeta potential is largely responsible for the suppressed diffusiophoretic particle motion in the presence of ATP. We note that while the zeta potential may, in principle, vary as the CMB moves through the nucleotide concentration field, the present

model yields accurate results for the drift without the need to account for this variation[50].

## Phoretic drift of CMB in a non-continuous flow system

Having established nucleotide-driven diffusiophoresis in the presence of continuous flow, we turn to a non-continuous flow setup, as shown in Fig. 3a. Similar setups have been previously used to study reaction-diffusion kinetics, spatially segregated chemistry, morphogenesis in synthetic systems, etc.[61–67]. Importantly, such a setup also allows for a longer experimental observation time, which generates greater phoretic drift. For the remainder of this work, we thus focus on diffusiophoresis (and later, catalytic activity) in a nucleotide gradient in the absence of continuous flow.

The experimental setup is shown in Fig. 3a, and consists of a centimetric chamber (length 35 mm, width 8.5 mm, height 0.05 mm), with two arms. First, we added nucleotide solution (1 mM) through the right arm and allowed it to fill the chamber completely. Then, (10 s after adding nucleotides), we added CMB-solution from the left arm of the centimetric chamber. We then tracked the concentrations of CMB in four zones (labeled A, B, C, and D) of length 1.5 mm each inside the chamber (Fig. 3a, details in *Methods*). After initial transients associated with the addition of the CMB, particles were observed to drift systematically up the gradient of nucleotide concentration. This motion occurred for a little over 5 minutes, at which time the CMB settled to the bottom of the glass chamber and stopped moving. Thus, we report the drift 5 min after the addition of CMB.

We first performed a control experiment with only milli-Q water (Supplementary Fig. 17, SI) and then with nucleotides. Particle motion was observed even in the control experiment with water, likely because of gravity and inertia associated with the process of adding them to the chamber. However, the CMB drift is greater in nucleotide solutions; Fig. 3c shows the excess drift in the presence of nucleotides (we

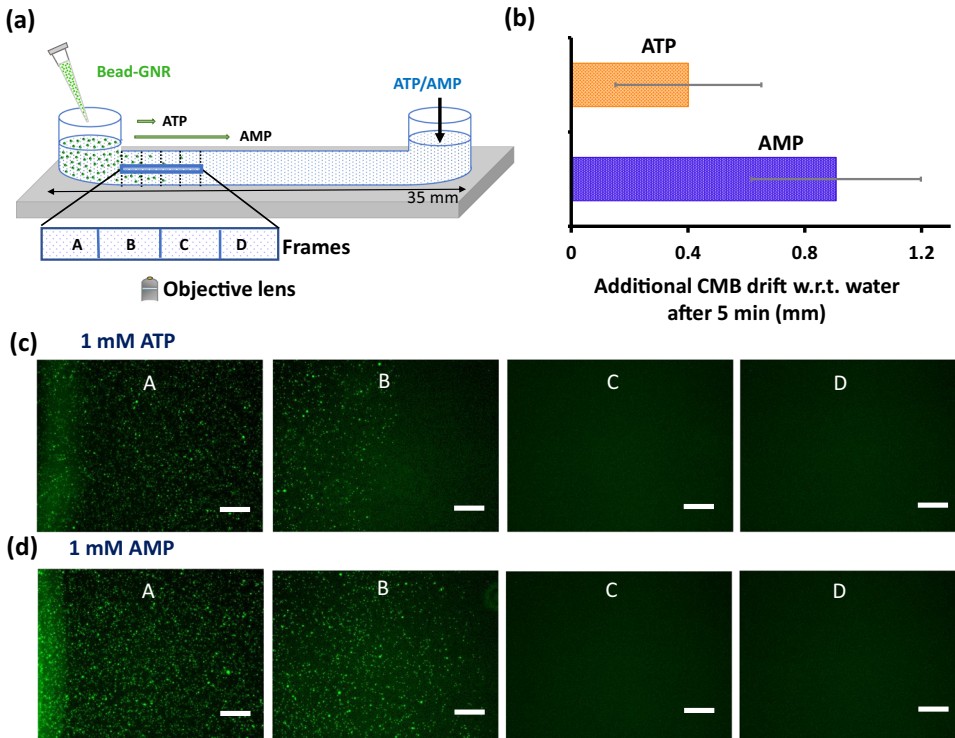

**Fig. 3 | Diffusiophoretic drift in a gradient of nucleotides without external flow. a** Schematic representation of the experimental setup. 50 μl of nucleotide solution with and without NBI solution was added from one end of the channel and 5 μl CMB solution was added from another end. **b** Additional distance covered by CMB when channel was filled with ATP or AMP (1 mM) with respect to the control (water only) after 5 min. Fluorescence images of the channel in different frames with (**c**) ATP and (**d**) AMP showing greater drift with AMP. The scale bar is 200 μm. Data are presented as mean ± SD (where $n = 5$). Source data are provided as a Source Data file.

subtract off the drift in the water-only control), which indicates the contribution of diffusiophoresis. Particles travel farther up the gradient of AMP compared to ATP by nearly 0.5 mm (see Fig. 3b, Supplementary Figs. 17, 18, SI). This is made evident by Fig. 3c, d, where particles covers the entirety of frame B with AMP, but only half of the same frame with ATP (each frame is 1.5 mm wide).

To understand the difference between AMP and ATP, we assume that the transport is controlled by the diffusion of nucleotides towards the left arm, which establishes gradients driving diffusiophoresis of CMB. We thus expect that the distance traveled by the CMB after time $t$ scales as $(2\Gamma_p t)^{1/2}$, where $\Gamma_p$ is the diffusiophoretic mobility, and is typically a fraction of the ambipolar diffusivity $D_a$ of the nucleotide (see SI for definitions). For AMP, $\Gamma_p \approx 5.7 \times 10^{-10}\,\mathrm{m^2/s}$, about ten times greater than in AMP (Supplementary Table 2). The estimated drift distance $(2\Gamma_p t)^{1/2}$ at the end of 5 minutes are on the order of a millimeter, consistent with experimental measurements. More importantly, the difference between AMP and ATP from this estimate is a little over 0.4 mm, which agrees well with the experiments (Fig. 3b). We additionally solve a one-dimensional numerical model of diffusiophoresis driven by the diffusion of nucleotides. Although the model is complicated by uncertainties associated with initial conditions (due to variability in the addition of the CMB), we find that it predicts that the CMB drift 0.3–0.4 mm more in the presence of AMP than with ATP, corroborating the scaling estimates, and in agreement with experiments (Fig. 3b, Supplementary Fig. 19, SI). This confirms that the diffusiophoresis, mediated by multivalency, is responsible for the differential transport of CMB in AMP versus ATP gradients in this non-continuous flow system.

## Phoretic jump of CMB during enzymatic ATP hydrolysis

We have already shown that multivalent interactions with ATP greatly suppress diffusiophoretic motion in ATP/AMP mixtures (Fig. 2g). We now demonstrate how this feature can be exploited to control the spatiotemporal organization of the CMB via in situ regulation of these multivalent interactions. To this end, we used potato apyrase enzyme (PA), which cleaves ATP to AMP + 2 Pi (phosphate ion), which is a strategy recently used to design programmed assemblies, reactors and catalysts[30,68,69].

We utilize the setup and protocol discussed earlier (Fig. 3a), but now use only 10 μM ATP, so that complete hydrolysis of ATP occurs within 5 min at a PA concentration of 100 nM (see SI for enzyme kinetics, Supplementary Figs. 24, 25). Thus, the hydrolysis reaction produces a solution containing $10\chi_{ATP}$ μM ATP + $10(1-\chi_{ATP})$ μM AMP + $20(1-\chi_{ATP})$ μM Pi, where the fraction of ATP, $\chi_{ATP}$ = [ATP]/([ATP] + [AMP]) decreases continually over time.

To establish the transport properties of the CMB in this system, we first performed control experiments (without the PA enzyme) with compositions of ATP + AMP + Pi of the type described above at different ATP fractions $\chi_{ATP}$. This mimics the chemical composition that would occur at different instants during the enzymatic reaction, but without the transients associated with the kinetics of this reaction. As before, we observed appreciable differences in the ζ-potential of CMB with 10 μM ATP ($\chi_{ATP} = 1$; ζ = 30 ± 3 mV) versus 10 μM AMP + 20 μM Pi ($\chi_{ATP} = 0$; ζ = 50 ± 2 mV) (Supplementary Fig. 20, SI). The additional drift of CMB (with respect to milliQ water only) observed at $\chi_{ATP}$ values of 0 and 1 are 3 ± 0.5 and 1.2 ± 0.3 mm (Supplementary Figs 21–23, SI). Interestingly, this drift remained roughly constant upon decreasing $\chi_{ATP}$ value from 1 to 0.4, and a sudden increase in drift was observed when decreasing $\chi_{ATP}$ below 0.3 (Supplementary Fig. 23, SI). This experiment exemplifies the role of the multivalent binding in driving diffusiophoretic effects, as the number of phosphate units (covalently linked for ATP or free phosphate for AMP + 2Pi) is constant in all cases.

We exploit this sensitivity of the phoresis to ATP to control the spatiotemporal distribution of CMB in situ, through the enzymatic hydrolysis of ATP. We filled the chamber with ATP (10 μM) and PA (100, 150, 200 and 250 nM) from one arm, and then added CMB solution

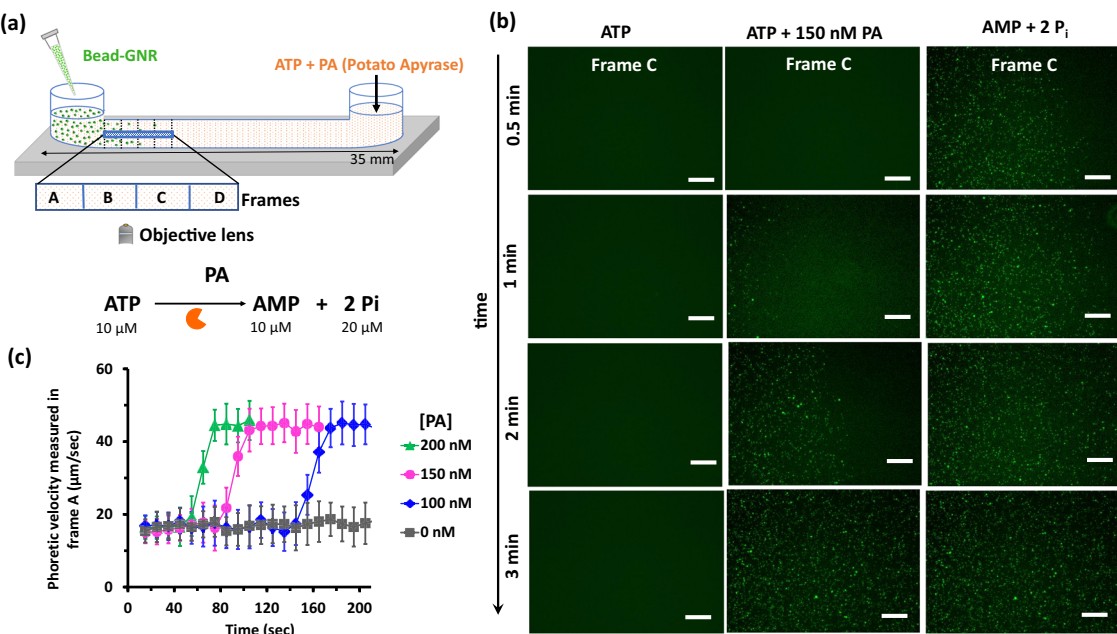

**Fig. 4 | Spatiotemporal evolution of colloidal population during enzymatic hydrolysis of ATP. a** Schematic representation of the experimental setup. ATP-containing potato apyrase was filled inside the channel and CMB solution was added from another end. **b** Fluorescence images in frame C over time when ATP or AMP + 2Pi or ATP + enzyme, potato apyrase (PA), (150 nM, ATP hydrolysis condition) was added inside channel indicate the population of CMB. **c** Phoretic velocity of CMB measured in frame A during ATP hydrolysis at different PA concentrations. Velocities were calculated from recorded videos. Experimental condition: 0.025 mg/mL bead, [GNR] = 37.5 pM, [ATP] =10 μM, [PA] = 100, 150, and 200 nM at 25 °C. Data are presented as mean ± SD (where $n$ = 3). Source data are provided as a source data file.

from the other arm (Fig. 4a). We then followed the particle population with time at frame C (about 4.5 mm from the source of the particles), which was completely devoid of particles over the experimental time in our control experiments (without PA) with ATP alone, but was densely populated with AMP + 2Pi. With 10 μM ATP and 150 nM PA, frame C started to become populated after only 1 min (Fig. 4b). Using 200 nM PA instead led to increase in CMB particle concentration in frame C observed after 1 min (Fig. S26, SI), confirming the role of ATP hydrolysis in controlling the motion of particles.

We also measured the diffusiophoretic velocity for the same experiment by tracking the particle motion in frame A. We average all trajectories across the frame, and use a 10 s moving-average window in time. The resulting velocity-versus-time data are plotted in Fig. 4c. In the presence of ATP but without PA, the drift velocity is roughly 15 ± 4 μm/s throughout our recording time of 200 s (Fig. 4c, supporting video 1, SI). In the presence of PA, the velocity was initially similar to the ATP-only case, but increases rapidly (i.e., "jumps") to over 40 μm/s after some time, where it remained constant for the rest of the experiment. With 100 and 150 nM PA, this jump in the phoretic velocity occurred around 75 and 140 seconds after the start of the experiment, respectively, whereas with 200 nM PA, the phoretic jump was observed after just 40 seconds (Fig. 4c, supporting video 2). At 250 nM PA, the phoretic velocity was found to be large (around 40 μm/s) right at the beginning of the experiment (supplementary Fig. 27, SI), suggesting that the jump must have occurred on timescales shorter than a few seconds. Our enzyme kinetic parameters (see SI) suggest that the observed rapid jump in phoretic velocity occurs after dissociation of around 70% ATP. These observations demonstrate that (i) the colloidal phoretic jump is controlled by the enzymatic downregulation of multivalent interactivity i.e. dissociation of ATP to AMP + 2Pi with time. (ii) the time at which the jump occurs can be controlled using the enzyme concentration, which modifies the reaction kinetics. Thus, we have shown that this enzymatic modulation of nucleotide cleavage can be used to control the population of colloids in both space and time.

## Controlling spatiotemporal catalysis by modulating the phoresis of CMB

We have thus far demonstrated one facet of the CMB's multivalent interaction with nucleotides, namely that they undergo diffusiophoretic motion in nucleotide gradients, and that this motion can be controlled enzymatically. A second facet of this interaction relates to the ability of this type of CMB to catalyze the proton transfer reaction (also known as Kemp elimination (KE)) in the presence of a phosphate buffer[53]. Furthermore, KE catalysis over a cationic gold nanoparticle surface is also controllable by modulating multivalent interactions with AMP, ADP, and ATP, as demonstrated in another study[54]. We combine both of these facets here, and achieve simultaneous in situ control of phoresis and catalysis, leading to spatiotemporal control of catalysis. Such control can not only have industrial importance in developing integrative catalysis with multiple reactions but also in generating non-equilibrium chemical reaction networks[1,61,70].

We characterized the ability of the CMB to catalyze the KE reaction by following UV-scanning kinetics using 5-nitrobenzisoxazole (NBI) as a substrate and tracking the change of absorbance of the product (2-cyano nitrophenol (CNP)) peak at 380 nm (Fig. 1; Supplementary Fig. 28a, SI). Carboxylate beads devoid of nanorods exhibit no catalytic activity (Supplementary Fig. 28b, SI). No catalysis was observed in the absence of nucleotides as the reaction follows the E2-pathway, which requires an anionic base (Supplementary Fig. 28c, SI)[53,54]. Interestingly, the addition of 1 mM AMP to the system enhances the reaction rate almost 1000-fold (Supplementary Fig. 29a, SI). The same amount of ADP leads to only a 10-fold enhancement of the reaction rate, while no catalysis occurs with ATP. In a mixed AMP/ATP system with the total concentration held constant at 1 mM (Supplementary Fig. 29b, SI), the KE catalytic ability decreased with increasing

ATP fraction $\chi_{ATP}$ = [ATP]/([ATP] + [AMP]), and is suppressed entirely when $\chi_{ATP}$ exceeds 0.5. This pattern of catalytic activity of CMBs in AMP/ATP mixtures is consistent with previous work involving cationic gold nanoparticles,[54] indicating that binding and nucleotide-mediated modulation of KE-catalysis persists on the cationic surface of the GNR even when it is bound to an anionic micron-sized polymer bead.

To exert simultaneous spatiotemporal control of catalysis and phoresis, we used an experimental setup (Fig. 5a, b) analogous to ones discussed previously, except for the addition of NBI along with nucleotides (AMP or ATP). We note that NBI is only utilized as the substrate of the KE reaction (which is catalyzed by CMB) and does not affect the diffusiophoresis of CMB (details in SI, Supplementary Figs. 30–32). We also included NaCl with the nucleotides in some experiments as a means to inhibit phoresis but not the KE reactivity (Fig. 5c, Supplementary Figs. 33–35, SI). We then monitored the phoresis of CMB (Fig. 5a–c, Supplementary Figs. 33, 34, SI) and the amount of KE product formed in Zone 1 and Zone 2 (indicated in Fig. 5b, Supplementary Fig. 36, SI) by measuring the absorbance with a plate reader.

Phoresis is suppressed with AMP + NaCl, and particles were absent from Zone 1 even after 5 min. With only AMP (and no NaCl), we observed KE product formation in Zone 1 almost immediately, while with AMP + NaCl, the reaction product was detected in Zone 1 a full 2 minutes after the start of the experiment. This suggests that with AMP + NaCl, where diffusiophoresis of CMB (the catalyst) is significantly suppressed, product formation occurred to the left of Zone 1 (see Fig. 5) and made a delayed appearance in Zone 1 due to diffusion of the reaction product. This observation exemplifies the role of diffusiophoresis in the spatial control of reaction products.

As discussed earlier, ATP inhibits both the phoresis and the KE reaction. However, enzymatically hydrolyzing ATP using PA generates AMP + 2Pi, simultaneously activating both the phoresis and the reaction. Using ATP in the presence of 500 nM PA, the concentration of product in Zone 1 started to increase after 1.5 min, associated with the production of sufficient AMP, activating both phoresis and the reaction. With a mixture of ATP + PA + NaCl, the product took over twice as long to form (Fig. 5d). This additional delay occurs because the NaCl suppresses phoresis of the CMB into Zone 1, but leaves their catalytic properties unchanged. These behaviors are qualitatively unchanged in Zone 2 which is farther away from the source of the CMB (Supplementary Fig. 36, SI).

Figure 5d summarizes a selection of these results, showing various strategies to control the location and first-appearance time of reaction products. These strategies employ in situ control of multivalent interactions, which simultaneously modulates the phoretic jump of microparticles (which determines the locations of the CMB) as well as the catalytic activity. The combination of these effects—both mediated by multivalent interactions of the CMB with nucleotides—can thus be tailored to tune the spatiotemporal organization of reaction products.

In conclusion, combining theoretical and experimental analyses, we showed how multivalent chemical fuel-driven interaction of catalytic microbeads with adenosine nucleotides modulate both the diffusiophoretic transport and chemical reactivity. The two key findings of this work relate to the transport and functionality of catalytic colloids: (i) the emergence of a colloidal phoretic jump, which is a rapid increase in phoretic speed that arises during in situ downregulation of multivalent interactivity and whose timing can be regulated by changing enzyme-mediated ATP hydrolysis, and (ii) spatial and temporal modulation of reaction products catalyzed by the colloids, by tuning phoresis and reactivity. Using these findings, we have demonstrated a route to spatiotemporally control the concentration of both colloids and the reaction products that they catalyze by fine-tuning their multivalent interactions with small, biologically relevant, molecules. This opens up new avenues to program the delivery of colloids for chemical

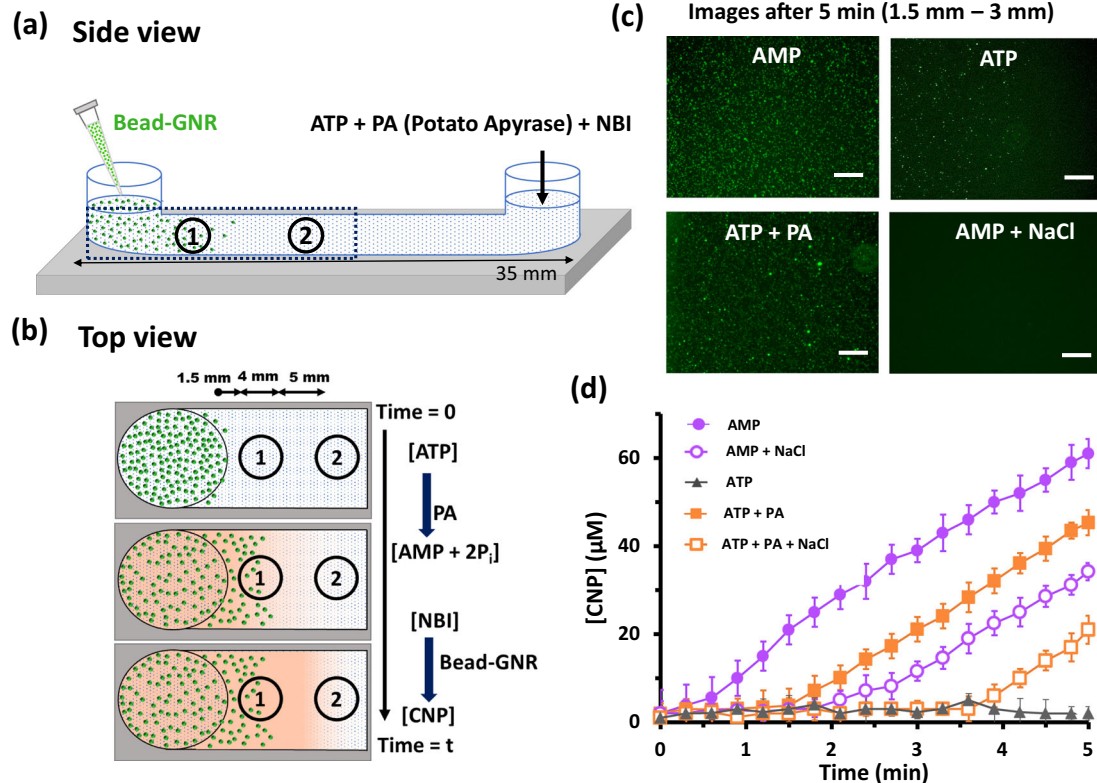

**Fig. 5 | Spatiotemporal control over catalysis by tuning phoresis of CMB.**
**a** Schematic representation of a channel filled with ATP, PA, and NBI from one arm and with Bead-GNR (CMB) solution added from another arm so that both ATP to AMP conversion and NBI to CNP conversion co-occur. The dotted rectangle marks the area of experimental observation where absorbance is recorded in Zone 1 and Zone 2. **b** Schematic of top view of the area of interest showing phoresis and conversion of both ATP and NBI inside channel over time. The distances between Zone 1 and the armhole and Zone 1 and Zone 2 are 1.5 mm, and 5 mm, respectively. The diameter of each zone is 4 mm. Orange shading denotes the distribution of reaction product (CNP) over time in the channel. **c** Microscopic images of CMB in the presence of different conditions are 1.5 mm distant from the armhole. Scale bar = 200 μm. **d** The amount of CNP formed in the presence of ATP only (no catalysis), AMP only (catalysis + phoresis), ATP + PA (enzymatic conversion and temporally controlled phoresis + catalysis), and AMP + NaCl (catalysis but no phoresis), and ATP + PA + NaCl (enzymatic conversion with weak phoresis) in Zone 1. Experimental condition: [AMP] = 0.1 mM, [ATP] = 0.1 mM, [NaCl] = 0.1 mM, [PA] = 500 nM. Data are presented as mean ± SD (where $n = 3$). Source data are provided as a source data file.

processes (e.g. catalysis or drug release[71–73]) as a function of space and time by switching on or off phoretic activity. The findings of this work thus enable precise control of particle transport in biological and other complex systems with reaction-diffusion processes.

## Methods

### Microfluidic set up and calculation of drift
We used a microfluidic set up with a two-inlet and one-outlet channel with dimensions of $17 \times 0.6 \times 0.1$ mm³ (length × width × height). We injected fluorescent CMB solution from one inlet and nucleotide (AMP (1 mM) or ATP (1 mM) or $\chi_{ATP}$ at 0.01, 0.1, 0.25, 0.5, 0.75) from the other, each at a flow rate of $Q/2 = 0.15$ ml/h, and observed the transverse drift of the CMB near the end of the channel (16 mm from the inlets merging) under a fluorescence microscope by scanning the zonal intensity (Fig. 2a). The intensity was normalized with respect to its integral across the channel, which is a measure of the number of particles per area and is a conserved quantity at steady state. The diffusiophoretic drift due to the presence of nucleotides was calculated by comparing the location where the normalized intensity is 0.1 (near the baseline) with that of the no-nucleotide condition i.e., only in the presence of milli-Q water. During each measurement, we ensured that the two flows at the inlet met almost exactly at the centerline of the channel (Supplementary Fig. 7, SI). Notably, the interface between the flows at the inlet deviates less than 3 μm from the centerline. This is much lower than the observed drift near the outlet (1.6 mm down-

stream apart) in case of nucleotides across multiple experimental trials (see Supplementary Figs. 8–16, SI).

### Theoretical calculation of phoretic drift
To understand the extent of drift in the experiments, we model the diffusiophoretic motion of particles in a co-flow geometry that mimics the experiment. We use a height-averaged description of the fluid flow and the transport of nucleotides and particles (Fig. 2e). Because the height of the channel $h$ is much smaller than its width $w$ in the experiments, the height-averaged velocity is approximately uniform across the channel and equals $u = Q/(wh)$[74]. The height-averaged nucleotide concentration $c(x,y,t)$ is governed by the advection-diffusion equation (written here for a single nucleotide)

$$\frac{\partial c}{\partial t} + u \frac{\partial c}{\partial x} = D\, \nabla^2 c, \tag{1}$$

where $D$ is the ambipolar diffusivity of the nucleotides. The concentration of nucleotides follows a step distribution at the inlet to the channel. Particles are transported by the flow and by diffusiophoresis due to gradients of the nucleotide, while also diffusing. The height-averaged concentration of particles, $n(x,y)$, is governed by

$$\frac{\partial n}{\partial t} + \nabla \cdot ((u\, \boldsymbol{e}_x + \boldsymbol{u}_{dp})n) = D_p \nabla^2 n, \tag{2}$$

where $\boldsymbol{u}_{dp} = \Gamma_p \nabla(\ln c)$ is the diffusiophoretic velocity of a particle, $\Gamma_p$ is its diffusiophoretic mobility, and $D_p$ is its diffusivity. The particle concentration is also subject to zero flux at the walls ($\partial n/\partial y = 0$ at $y = \pm w/2$) and a step distribution of concentration at the inlet $n(x = 0, y, t) = n_0 \Theta(w/2 - y)$, with $n_0$ representing the particle concentration in the bottom inlet. We calculate the diffusiophoretic mobility $\Gamma_p$ and the ambipolar diffusivity $D$ by modeling the nucleotide in solution as a salt involving a mono- di- or tri-anion for AMP, ADP, and AMP, respectively, with the appropriate number (one, two or three, respectively) of Na$^+$ cations. We emphasize that $\Gamma_p$ depends on the zeta-potential of the particle (which we take directly from experimental measurements) and comprises a combination of an electrophoretic component and a chemiphoretic component; see SI.

We solve Eqs. (1)–(2) numerically after a suitable non-dimensionalization. The nucleotides advect along and diffuse across the channel producing concentration gradients that drive the phoretic motion of the particles. We run the model until a steady state is established by the balance of advective, diffusiophoretic and diffusive fluxes. To compare with experiments, we report steady-state nucleotide and particle concentrations near the end of the channel ($L = 1.6$ mm from the inlet; cf. Fig. 2). Figure 2e shows lateral profiles of normalized particle concentrations at the end of the channel. The normalization reflects the fact that the number of particles at any cross-section along the flow is constant. We find that the positively charged particles drift diffusiophoretically towards the nucleotide, consistent with the experiments. As seen in the experiments, AMP produces the greatest drift, followed by ADP, and then followed by ATP, which produces only a very weak drift. The extent of drift is controlled primarily by the differing ζ-potential for these different nucleotides, with the differing ionic charge and diffusivities of the nucleotides playing a secondary role. We follow the experimental protocol and compute the drift as the $y$-location where the normalized concentration profiles attain a value of 0.1 at a downstream location $x = 1.6$ cm. We follow a similar strategy for a mixture of nucleotides, as described in the main text; see also SI.

### CMB population study in the non-continuous flow setup
Our experimental setup for non-continuous flow is shown in Fig. 3a. In all cases, we performed measurements up to 5 min as after that time, we observed settling of the particles at the bottom of the glass channel (Supplementary Fig. 21, SI). Here, we added 50 µl of either ATP or AMP or AMP + NBI through one arm and filled the chamber having dimensions $35 \times 8.5 \times 0.05$ mm$^3$ (length × width × height). The concentrations of nucleotides were maintained at 1 mM as in the microfluidic study. Next, we added 5 µl of the CMB (0.025 mg/ml bead, 5 times more dilute than the concentration used in the microfluidic study) and followed the particle motion inside the chamber using the microscope. We followed the population dynamics at four zones, namely frames A, B, C, and D, each having a length of 1.5 mm. During the enzymatic reaction, the channel was filled with nucleotide solution (with or without PA and NBI) through one arm, allowed to settle for 10 sec and then CMB was added from another arm. The instant at which CMB is added is taken to be the origin of time in the experiment. The population and phoretic velocity of CMB in the channel were monitored using a fluorescence microscope. Notably, Ca$^{2+}$ in the form of Ca(NO$_3$)$_2$ (0.25 mM) was added for the activation of enzyme, PA (for Figs. 4, 5). Here, Ca$^{2+}$ was present in both bead-GNR (CMB) solution and (ATP + PA) or (ATP + PA + NBI) mixture to nullify gradient formation by Ca-salt.

### Diffusiophoretic velocity measurement
For extracting the velocity of CMB (Bead-GNR conjugate) from experimental videos Zen 3.0 (blue edition) software was used. For this, the displacement covered by CMB was measured for every 10 seconds. Every point in the velocity is the average within a 10 sec window. For example, the point denoted as 15 and 25 s, is the average velocity of

10–20 and 21–30 s, respectively. Zero point in time is considered after the addition of CMB.

## Data availability
The authors declare that the data supporting the findings of this study are available within the paper and its Supplementary Information. Source data are provided in this paper. Source data are provided with this paper.

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

## Acknowledgements

S.M. acknowledges financial support of Science and Engineering Research Board (SERB) (File No. CRG/2022/002345). E.S. acknowledges CSIR, India (09/947(0109)/2019-EMR-I) for doctoral research grant.

## Author contributions

E.S. performed the experiments. B.R. carried out the theory. S.M. conceived the idea and supervised the experimental work. All the authors prepared the manuscript and supplementary information.

## Competing interests

The authors declare no competing interests.
