## [Peer Review File · Nature Communications]

In situ enzymatic control of colloidal phoresis and catalysis through hydrolysis of ATPREVIEWER COMMENTS

Reviewer #1 (Remarks to the Author):

The authors combine features of colloid science and molecular biology to demonstrate movement of colloidal particles in a chemical gradient, where the particles are specially coated to be responsive to the chosen chemistry. They show how multivalent interactions of adenosine nucleotides (AMP, ADP, ATP) – so, respectively, monovalent, divalent, and trivalent charges - with catalytic microbeads produce directed transport of the beads. The nucleotides bind to the coated particle surface with ATP binding the strongest. I believe they track a chemical reaction (Kemp elimination) though I was unclear on this point. The change in chemical kinetics alone apparently was reported already by the authors (reference 43 I think) so the novelty here is showing that the kinetics can be coupled to particle transport, which seems original and nicely done and will be of interest to readers. The kinetics is such that enzymatic downregulation of multivalent activity via ATP hydrolysis to AMP gives control over particle drift, which causes a rapid increase in transport, so that the authors refer to it as a phoretic leap. In particular, as might be guessed from the description above where ATP has preferential binding to the catalytic surface, multivalent interactions with ATP greatly suppress diffusiophoretic motion in ATP/AMP mixtures. They then show that they can achieve additional control by using ‘in situ’ degradation of the multivalent interactivity with the potato apyrase enzyme (PA), which cleaves ATP to AMP+ 2 phosphate groups. The configuration of Figure 2 shows a true steady but figures 3 and 4 in my understanding yield transient configurations but they are described in the text with a “steady state language”, which I find confusing, e.g., velocities are stated for figures 3 and 4 aren’t all the gradients changing in time (so would all the particle speeds)? Overall, the diffusiophoretic transport of particles has largely been confined to the colloid science literature. One recent paper (reference 41) demonstrates the active diffusiophoretic response in a biological system. In this paper, the two topics – colloid science with kinetic control from biology – are linked and the modeling of the diffusiophoretic transport impressively captures the experimental trend, including that of the effect of the different valences present. I provide some questions for clarification below but I think this paper is among the first to really link the chemical control capabilities based on biology (here AMP, ADP, ATP) with directed transport based on diffusiophoretic motion of colloidal particles. I

believe readers will find this link interesting at it will suggest new avenues of research.

Additional remarks:

1) p. 7-8: What is the feature that controls drift in mixtures of AMP/ADP/ATP? Diffusivity of the molecules? Charge of the molecules? Different interactions with the substrate of the particles? This did not seem clear although when comparing theory and experiments the authors write that there are no fitting parameters.

2) p. 9: cleaves ATP to AMP + 2 Pi -is a phosphate denoted Pi?

3) Fig 3: “macroscale” versus “microfluidic” – I do not see why these are thought to be different. The motions are laminar, the driving force diffusive, etc. The biggest difference in my view is that figure 2 allows a steady state to be achieved but figures 3 and 4 look to me like they will always have a transient response.

4) P 3: “simultaneous control of chemistry and transport – has not been investigated in previously researched systems”. I think this is likely true if interpreted narrowly but studies of the change of surface chemistry that are tied to pH have been reported and this is a form of transport and reaction: Shim et al. Diffusiophoresis in the presence of a pH gradient" Phys. Rev. Fluids 2022, 7, 110513.

5) p. 6: It seems odd to me to write an absolute number without any other sense of distances (e.g., typically width, how far down stream, etc.): “Particles drift toward the nucleotide in the cases of AMP and ADP by 63 ± 4 and 57 ± 3 μm , respectively, and by much less (only about 13 ± 5 μm) in ATP gradients”. This occurs again on p. 10: “The additional drift of CMB (with respect to milliQ water only) observed at χ_{ATP} values of 0 and 1 are 3 ± 0.5 and 1.2 ± 0.3 mm”. Perhaps there is some more systematic way to report the quantitative features that have a little more generality.

6) In the Methods, there is a theoretical model. What does it mean to have a “steady state” but the equation (1) still has a time derivative? Then we are told about “ $n(x,y)$ ” and directed

to equation (2) which includes a time derivative so I assume $n(x,y,t)$?

7) p. 16: "we report steady-state nucleotide and particle concentrations near the end of the channel." Are these quantities changing in time or are they "steady states"?

8) The title sounds funny to me ("altering multivalent ...")

9) There are clearly spatiotemporal effects and some control but perhaps it is less strong than hinted in the title.

Other: The article is filled with various misprints or poor proofreading

a) Abstract: correct spelling "diminished"; p. 3: design of drug molecules; p. 3: been controlled in-situ

b) p. 3: bound with a cationic cetyltrimethylammonium bromide (CTAB)-coated gold nanorods (GNR)

c) Sometimes they write "physicochemical" and other times "physico-chemical"

d) Abbreviation looks strange: Kemp elimination product (CNP)? But then the authors also write "2-cyano nitrophenol (CNP))"?

e) Figure 2 caption: "the drift of the the fluorescent CMB" and moreover the grammar is poor: "showing that the drift of the the fluorescent CMB towards AMP greater than ATP and control."

f) p. 7: "decreasing from"

g) p. 8: "well-established theory for multivalent mixtures"; "where we observed that"

h) include the abbreviation of PA in the caption to figure 4.

i) p. 11: “over the experimental time in in our control experiments”

j) p. 12: “simultaensouly activating”

k) figure 5 caption: “Orange shed” -> do the authors mean “orange shade”?

Reviewer #2 (Remarks to the Author):

There is too much jargon for this paper to even be readable, with reference to enzymes leaping (they seem to mean a sharp increase in drift velocity after some period of time. This is an interesting observation illustrated in Fig. 4 c, but I cannot penetrate what is actually being done. What does zero time denote? Why do the authors not look at the lag for 100 nM PA and 250 nM PA. It seems that the claim is that PA influences the lag time, but not the maximum velocity, but two points are not sufficient to make this point.

What is an "altering multivalent interaction field" that appears in the title, it is not explained in the text and I cannot come up with a sensible hypothesis as to what this combination of words might mean.

It is also strange that there are no citations to the work of major contributors in the field of active colloid motility such as Ayusman Sen, Ramin Golestanian, Steve Granick, Anna Balazs, Zarazar, Beves, etc., nor is there any mention of the active matter community.

I have the feeling that there is something interesting in this paper, but the obscure language, jargon, and overall unclear writing makes it almost impossible to uncover. This paper could possibly be rewritten with a view to clarity that would allow readers to see what is important, but that would require, in my opinion, a very major revision.

Reviewer #3 (Remarks to the Author):

The paper describes interested experimental observations that seem to be in line with our current understanding of diffusiophoresis. The authors explore a system where ATP

conversion to ADP and AMP results in simultaneous strong effects regarding phoretic motion. The strong effects can be understood by the significant different mobilities that each compound presents for the particle phoresis.

The subtle and biological relevant dynamics that can be found in this system are interesting and worth publication. However, I feel that the authors could improve the explanation, especially quantitative, regarding the observed phenomena. For instance, drift distances are provided, as opposed to determining the phoretic mobilities. Theories are available, based on particle zeta potential, to predict the mobility based on chemo and electrophoretic contributions. These are provided in the SI, but no comparison has been made based on the observed drift distances/velocities in the other experiments (figure 3 and 4). Perhaps it is the difficulty in predicting/quantifying the concentration gradient during the experiments.

For ionic solutes, the phoretic velocity scales with the relative concentration gradient. This seems crucial for the time evolution of this velocity in a spatially changing concentration gradient.

The microfluidic designs seem to pose some experimental challenges that the authors do not report on. For the drift experiments (figure 2) it will be very difficult to maintain the interface between the two solutions at exact position. Slight differences in flow rate affect the middle position and with that the determination of the drift distance. For the other experiments (figure 3 and 4), the establishment of the concentration gradient is not straightforward. Filling of the channel from one side, followed by filling the other reservoir for sure introduces some advective flow within the device. How has this been minimalized, or quantified? The initial situation is crucial, as the concentration gradients at that moment determine the phoresis, but also set the spatial concentration gradients that follow.

The authors have ignored any diffusio osmotic flow caused by the gradient along the channel wall. This flow can also affect the particle motion and is not based on their phoresis. At least, the magnitude of this flow should be established.

RESPONSE TO REVIEWERS' COMMENTS

Reviewer #1 (Remarks to the Author):

The authors combine features of colloid science and molecular biology to demonstrate movement of colloidal particles in a chemical gradient, where the particles are specially coated to be responsive to the chosen chemistry. They show how multivalent interactions of adenosine nucleotides (AMP, ADP, ATP) – so, respectively, monovalent, divalent, and trivalent charges - with catalytic microbeads produce directed transport of the beads. The nucleotides bind to the coated particle surface with ATP binding the strongest. I believe they track a chemical reaction (Kemp elimination) though I was unclear on this point. The change in chemical kinetics alone apparently was reported already by the authors (reference 43 I think) so the novelty here is showing that the kinetics can be coupled to particle transport, which seems original and nicely done and will be of interest to readers.

- We thank the reviewer for finding the work original and of potential interest to readers.

The kinetics is such that enzymatic downregulation of multivalent activity via ATP hydrolysis to AMP gives control over particle drift, which causes a rapid increase in transport, so that the authors refer to it as a phoretic leap. In particular, as might be guessed from the description above where ATP has preferential binding to the catalytic surface, multivalent interactions with ATP greatly suppress diffusiophoretic motion in ATP/AMP mixtures. They then show that they can achieve additional control by using ‘in situ’ degradation of the multivalent interactivity with the potato apyrase enzyme (PA), which cleaves ATP to AMP+ 2 phosphate groups. The configuration of Figure 2 shows a true steady but figures 3 and 4 in my understanding yield transient configurations but they are described in the text with a “steady state language”, which I find confusing, e.g., velocities are stated for figures 3 and 4 aren't all the gradients changing in time (so would all the particle speeds)? Overall, the diffusiophoretic transport of particles has largely been confined to the colloid science literature. One recent paper (reference 41) demonstrates the active diffusiophoretic response in a biological system. In this paper, the two topics – colloid science with kinetic control from biology – are linked and the modeling of the diffusiophoretic transport impressively captures the experimental trend, including that of the effect of the different valences present. I provide some questions for clarification below but I think this paper is among the first to really link the chemical control capabilities based on biology (here AMP, ADP, ATP) with directed transport based on diffusiophoretic motion of colloidal particles. I believe readers will find this link interesting at it will suggest new avenues of research.

- We thank the reviewer for encouraging comments, such as:
‘this paper is among the first to really link the chemical control capabilities based on biology (here AMP, ADP, ATP) with directed transport based on diffusiophoretic motion of colloidal

particles. I believe readers will find this link interesting as it will suggest new avenues of research.’

Additional remarks:

1) p. 7-8: What is the feature that controls drift in mixtures of AMP/ADP/ATP? Diffusivity of the molecules? Charge of the molecules? Different interactions with the substrate of the particles? This did not seem clear although when comparing theory and experiments the authors write that there are no fitting parameters.

- In a nucleotide mixture, the drift is governed primarily by the change in zeta potential, which is regulated by the individual binding affinity of nucleotides. We have shown here that the effectiveness of binding between the bead and nucleotide increases with increasing valency. Due to multivalent interactions (simultaneous interaction with multiple charged phosphates, e.g. ATP with the bead) ATP binds much stronger than AMP. Therefore, during an ATP/AMP mixture experiment, the zeta potential of the surface is dictated by ATP rather AMP, even at a lower mole fraction of ATP (please see supplementary figure 5). We note that selective binding of higher valent nucleotides even in the presence of lower valent nucleotides on cationic nanoparticle surfaces has previously reported by us (Ref. 54) and others (e.g. Cristian Pezzato, Paolo Scrimin, Leonard J. Prins, *Angew. Chem. Int. Ed.* 2014, 53, 2104-2109). We include the following lines in page 8 of the revised manuscript -

“We repeated the experiments described above with a 1mM mixture of AMP and ATP nucleotides, and studied the drift distance as a function of the fraction of ATP, $\chi_{ATP} = [ATP]/([ATP] + [AMP])$ (Supplementary Fig. 12-16, Supplementary Table 4, SI). The drift was about 63 μm in AMP ($\chi_{ATP} = 0$) as noted previously, and decreased sharply with the addition of ATP, dropping by over 50% with as little as 10 μM ATP (the mixture also contains 990 μM of AMP; $\chi_{ATP} = 0.01$), and down to 20 μm at $\chi_{ATP} = 0.25$. These findings are reproduced with excellent quantitative accuracy by the modeling framework, again with no fitting parameters (Figure 2g). As before, we use experimentally measured zeta potentials (see supplementary Fig. 6 for these measurements) as inputs to the model, but we now track the transport of individual nucleotides. We then relate their concentration gradients to the particle motion using the well-established diffusiophoretic theory for multivalent mixtures (see Supplementary Information), again accounting for both electrophoretic and chemiphoretic contributions.^{33,39}

These observations clearly demonstrate that a small fraction of ATP greatly suppresses the AMP-mediated phoretic motion of the CMB. ATP binds much more strongly than AMP due to multivalent interactions (simultaneous interaction of multiple charged phosphates with bead); similar selective binding of higher valent nucleotides on cationic nanoparticle surfaces, even in the presence

of lower valent nucleotides, has been reported in other systems.^{22,54} Therefore, during an ATP/AMP mixture experiment, the zeta potential of the surface is strongly controlled by ATP, even at relatively low mole fractions. The model confirms that this lowered zeta potential is largely responsible for the suppressed diffusiophoretic particle motion in the presence of ATP.”

2) p. 9: cleaves ATP to AMP + 2 Pi -is a phosphate denoted Pi?.

- Yes, the free phosphate ion is designated as Pi. This is noted in the revised manuscript.

3) Fig 3: “macroscale” versus “microfluidic” – I do not see why these are thought to be different. The motions are laminar, the driving force diffusive, etc. The biggest difference in my view is that figure 2 allows a steady state to be achieved but figures 3 and 4 look to me like they will always have a transient response.

- We agree with the reviewer regarding this issue. In the revised manuscript, we removed the term ‘macroscale’ rather we use the term ‘**non-continuous flow set up**’.

In figure 3 and 4, we were not able to follow the kinetics for more than 5 min as the particle started to settle at the bottom of the glass chamber and stopped moving. We added the following description of the detailed experimental protocol in the revised manuscript:

“The experimental setup is shown in Fig. 3a, and consists of a centimetric chamber (length 35 mm, width 8.5 mm, height 0.05 mm), with two arms. First, we added nucleotide solution (1 mM) through the right arm and allowed it to fill the chamber completely. Then, (10 sec after adding nucleotides), we added CMB-solution from the left arm of the centimetric chamber. We then tracked the concentrations of CMB in four zones (labeled A, B, C, and D) of length 1.5 mm each inside the chamber (Fig. 3a, details in Methods). After initial transients associated with the addition of the CMB, particles were observed to drift systematically up the gradient of nucleotide concentration. This motion occurred for a little over 5 minutes, at which time the CMB settled to the bottom of the glass chamber and stopped moving. Thus, we report the drift 5 minutes after the addition of CMB.”

4) P 3: “simultaneous control of chemistry and transport – has not been investigated in previously researched systems”. I think this is likely true if interpreted narrowly but studies of the change of surface chemistry that are tied to pH have been reported and this is a form of transport and reaction: Shim et al. Diffusiophoresis in the presence of a pH gradient" Phys. Rev. Fluids 2022, 7, 110513.

- We thank the reviewer for pointing this. In the revised manuscript, we incorporated the reference. Please see reference no. 38 in the revised version of the manuscript.

5) p. 6: It seems odd to me to write an absolute number without any other sense of distances (e.g., typically width, how far down stream, etc.): “Particles drift toward the nucleotide in the cases of AMP and ADP by 63 ± 4 and 57 ± 3 μm , respectively, and by much less (only about 13 ± 5 μm) in ATP gradients”. This occurs again on p. 10: “The additional drift of CMB (with respect to milliQ water only) observed at χATP values of 0 and 1 are 3 ± 0.5 and 1.2 ± 0.3 mm”. Perhaps there is some more systematic way to report the quantitative features that have a little more generality.

- We realize on reading the referee’s remark that we had relegated too much experimental detail into the supplementary material of the original manuscript. We have corrected this in the revised manuscript, which now includes the following text:

“We set up a microfluidic experiment with a two-inlet and one-outlet channel of width $w= 600$ μm and height $h= 100$ μm , while injecting CMB and 1 mM nucleotide solution into two inlets, each at a flow rate of $Q/2=150$ $\mu\text{l/h}$ (Fig. 2a, details in Methods). The CMB were observed to drift across the channel towards the nucleotides as they flow through the channel, while no such drift was observed in the absence of nucleotides (Fig. 2b). We measured the concentration profiles of CMB across the channel width, at a location 1.6 mm downstream of the inlets, and then normalized these concentrations by their integral across the channel, reflecting the fact that the number of particles at any cross-section remains constant (Fig. 2c). We define the cross-stream drift distance as the location at which the normalized particle concentration profiles attain a value of 0.1 (under this definition, the maximum possible drift distance is the half-width of the channel, 300 μm); see Fig. 2c. We find that particles drift toward the nucleotide (up the gradient) in the cases of AMP and ADP by 63 ± 4 and 57 ± 3 μm , respectively, and by much less (only about 13 ± 5 μm) with ATP (Fig. 2e, supplementary fig. 8-11, supplementary Table 3). These significant drift distances (as large as 21% of the half-width of the channel for AMP) are much greater than the variability from trial to trial (see Supplementary information).”

In a similar way, for the non-continuous flow set up case, we also mentioned the channel dimension in the captions of Figures 3, 4 and 5.

“The experimental setup is shown in Fig. 3a, and consists of a centimetric chamber (length 35 mm, width 8.5 mm, height 0.05 mm), with two arms.”

6) In the Methods, there is a theoretical model. What does it mean to have a “steady state” but the equation (1) still has a time derivative? Then we are told about “ $n(x,y)$ ” and directed to equation (2) which includes a time derivative so I assume $n(x,y,t)$?

- The model is formulated in a time-dependent framework for generality and for ease of computation, but we run it to steady state to compare with experiments. This steady state is established by a balance of advective (along the channel) and diffusive (across the channel) fluxes for the

nucleotides, and a balance of advective, diffusiophoretic and (to lesser extent) diffusive fluxes for the colloidal particles. We note that the experiments described in Fig. 2 are at steady state, as are the model results. We have clarified the language around Equation (1) to reflect these ideas.

The revised manuscript also includes a theoretical estimate in the non-continuous flow setup, where a steady state is not achieved. Here the time-dependence is indeed important.

7) p. 16: “we report steady-state nucleotide and particle concentrations near the end of the channel.” Are these quantities changing in time or are they “steady states”?

- As noted above, both nucleotide and particle concentrations are at a true steady state. This steady state corresponds to solutions of Eq. (1) and (2) with no time-dependent terms having relaxed to zero (this is a well-defined limit for the continuous flow system). In practice it is more straightforward to solve the time-dependent problem until the steady state is reached (which we check numerically). The steady state is reached roughly at a timescale that is the longer of an advective timescale L/u and a diffusive timescale w^2/D .

8) The title sounds funny to me (“altering multivalent ...”)

- We agree that ‘altering multivalent’ is ambiguous and we have removed it from the title. The revised title is:
“In situ enzymatic control of colloidal phoresis and catalysis through hydrolysis of ATP”

9) There are clearly spatiotemporal effects and some control but perhaps it is less strong than hinted in the title.

- We thought about this point and agree with the reviewer. The revised title is: “In situ enzymatic control of colloidal phoresis and catalysis through hydrolysis of ATP”

Other: The article is filled with various misprints or poor proofreading

- We are thankful to the reviewer for finding all the errors. We have corrected them in the revision.

a) Abstract: correct spelling “diminished”; p. 3: design of drug molecules; p. 3: been controlled in-situ

- We have corrected this and other errors.

b) p. 3: bound with a cationic cetyltrimethylammonium bromide (CTAB)-coated gold nanorods (GNR)

- We corrected this by removing ‘a’.

c) Sometimes they write “physicochemical” and other times “physico-chemical”

- We changed all appearances of these phrases to 'physicochemical'.

d) Abbreviation looks strange: Kemp elimination product (CNP)? But then the authors also write "2-cyano nitrophenol (CNP))"?

- In the very first instance where we are mentioning Kemp elimination product, we introduced 2-cyano nitrophenol (Fig. 1 caption) and abbreviated it as CNP.

e) Figure 2 caption: "the drift of the the fluorescent CMB" and moreover the grammar is poor: "showing that the drift of the the fluorescent CMB towards AMP greater than ATP and control."

- We corrected this error.

f) p. 7: "dcreasing from"

- This has been corrected.

g) p. 8: "well-established theory for multivalent mixtures"; "where we observed that"

- We corrected the spelling of 'multivalent' and 'observed'.

h) include the abbreviation of PA in the caption to figure 4.

- We included the abbreviation of PA as potato apyrase.

i) p. 11: "over the experimental time in in our control experiments"

- We corrected this by removing the extra 'in'.

j) p. 12: "simultaensouly activating"

- We corrected the spelling of 'simultaneously'.

k) figure 5 caption: "Orange shed" -> do the authors mean "orange shade"?

- Yes, 'orange shade'. We have corrected this error.

Reviewer #2 (Remarks to the Author):

There is too much jargon for this paper to even be readable, with reference to enzymes leaping (they seem to mean a sharp increase in drift velocity after some period of time. This is an interesting observation illustrated in Fig. 4 c, but I cannot penetrate what is actually being done. What does zero time denote? Why do the authors not look at the lag for 100 nM PA and 250 nM PA. It seems that the claim is that PA influences the lag time, but not the maximum velocity, but two points are not sufficient to make this point.

- We thank the reviewer for pointing out these issues with the original submission.

We agree that the original manuscript was likely too condensed and had too much jargon. We therefore undertook a significant rewriting of the manuscript, and believe that the revised version is significantly more organized and approachable to the reader. We have added several additional textual and experimental details in the main manuscript to make it more readable and understandable.

To the referee's point about varying PA concentrations, we also performed the experiments with 100 and 250 nM PA. At 100 nM PA, the phoretic jump (which we previously called a "leap") takes longer to occur. At 250 nM PA, the jump occurs so early (likely on the timescale of a few seconds) that we cannot identify it as such, since the velocity is already at the higher value of about 40 $\mu\text{m/s}$ at the start of the measurements.

We include below the revised Figure 4 and Figure S27 in SI.

In the main manuscript we added the following text and modified the figure captions to denote the zero time in our experimental procedure. The zero point in time is identified as the time at which the CMB beads were added, exactly 10 sec after the addition of (ATP + PA) solution in the reservoir through the other arm of the channel.

Fig. 4. Spatiotemporal evolution of colloidal population during enzymatic hydrolysis of ATP. (a) Schematic representation of the experimental setup. ATP containing potato apyrase was filled inside the channel and CMB solution was added from another end. (b) Fluorescence images in frame C over time when ATP or AMP + 2Pi or ATP + enzyme, potato apyrase (PA), (150 nM, ATP hydrolysis condition) was added inside channel indicate the population of CMB. (c) Phoretic velocity of CMB measured in frame A during ATP hydrolysis at different PA concentration. Velocities were calculated from recorded videos. Experimental condition: 0.025 mg/mL bead, [GNR] = 37.5 pM, [ATP] = 10 μM, [PA] = 100, 150 and 200 nM at 25 °C. Error bars are standard deviations of 3 independent experiments.

With 100 and 150 nM PA, this jump of the phoretic velocity occurred around 75 and 140 seconds after the start of the experiment, respectively, whereas with 200 nM PA, the phoretic jump was observed after just 40 seconds (Figure 4c, supporting video 2). At 250 nM PA, the phoretic velocity was found to be large (around 40 μm/s) right at the beginning of the experiment (supplementary Fig. 27, SI), suggesting that the jump must have occurred on timescales shorter than a few seconds. Our enzyme kinetic parameters (see SI) suggest that the observed rapid jump in phoretic velocity occurs after dissociation of around 70% ATP. These observations demonstrate that (i) the colloidal phoretic jump is controlled by the enzymatic downregulation of multivalent interactivity i.e. dissociation of ATP to AMP + 2Pi with time. (ii) the time at which the jump occurs can be controlled using the enzyme concentration, which modifies the reaction kinetics. Thus, we have shown that this enzymatic modulation of nucleotide cleavage can be used to control the population of colloids in both space and time.

Supplementary Fig. 27. Phoretic velocity of CMB measured in frame A during ATP hydrolysis at 250 nM PA concentration. Please see Figure 4 of the main manuscript for experimental set up. Experimental condition: 0.025 mg/mL bead, [GNR] = 37.5 pM, [ATP] = 10 μM, [PA] = 250 nM at 25 °C. Channel length = 35 mm, height = 0.05 mm and width = 8.5 mm. The error bar is standard deviation of 2 independent experiments.

Additionally, we added the following statement in the revised “Methods” section:

“For extracting the velocity of CMB (Bead-GNR conjugate) from experimental videos Zen 3.0 (blue edition) software was used. For this, the displacement covered by CMB was measured for every 10 seconds. Every point in the velocity is the average of 10 sec. For example, the point denoted as 15 and 25 sec, is the average velocity of 10-20 and 21-30 sec, respectively. **Here zero point in time is calculated after the addition of CMB beads.**”

What is an "altering multivalent interaction field" that appears in the title, it is not explained in the text and I cannot come up with a sensible hypothesis as to what this combination of words might mean.

- We agree with the reviewer and thank them for raising this issue. In the revised manuscript, we changed the title to: “In situ enzymatic control of colloidal phoresis and catalysis through hydrolysis of ATP”, which according to us is more transparent and more accurate.

Our original intent was to use “altering multivalent interaction” to refer to the binding of ATP containing 3-covalently attached phosphate with CMB. In this work, we degrade the multivalency of ATP by enzymatically converting it to AMP + 2Pi, which simultaneously affects the binding interaction of CMB with nucleotides over time. Furthermore, selective binding of higher valent nucleotides even in presence of lower valent nucleotides on cationic nanoparticle surface has previously reported by us (Ref. 52) and others (for example: Cristian Pezzato, Paolo Scrimin, Leonard J. Prins, *Angew. Chem. Int. Ed.* 2014, 53, 2104-2109).

We agree that these facets are not adequately captured by the phrase “altering multivalent interaction field” so we have changed the title accordingly. Nonetheless, these are central to the present work, and are discussed at length in the revised manuscript.

It is also strange that there are no citations to the work of major contributors in the field of active colloid motility such as Ayusman Sen, Ramin Golestanian, Steve Granick, Anna Balazs, Zarazar, Beves, etc., nor is there any mention of the active matter community.

- We thank the reviewer for this suggestion. In the introduction and throughout the text of the revised manuscript, we cite recent work by many of these authors and more. For example, the introduction now includes the sentence: added the following sentence, “*Recent work has sought to understand and control the transport of colloidal objects – ranging from nanometer-sized enzymes to the micron-sized droplets and polymeric beads – in a variety of chemical gradients.* 4-11”
- The newly added references are following:
 4. Kar, A., Chiang, T. -Y., Riviera, I. O., Sen, A. & Velegol, D. Enhanced Transport into and out of Dead -End Pores. ACS Nano 9, 746–753 (2015).
 5. Daddi, M. -I. A., Vilfan, A. & Golestanian, R. Diffusiophoretic propulsion of an isotropic active colloidal particle near a finite-sized disk embedded in a planar fluid–fluid interface. J. Fluid Mech. 940, A12 (2022).
 6. Illien, P., Golestanian, R. & Sen, A. ‘Fuelled’ motion: phoretic motility and collective behaviour of active colloids. Chem. Soc. Rev. 46, 5508-5518 (2017).
 7. Agudo-Canalejo, J., Illien, P.& Golestanian, R. Phoresis and Enhanced Diffusion Compete in Enzyme Chemotaxis. Nano Lett. 18, 2711-2717 (2018).
 8. Laskar, A., Shkylaev, O. & Balazs, A. Collaboration and competition between active sheets for self-propelled particles. Proc. Natl. Acad. Sci. U.S.A. 116, 9257-9262 (2019).
 9. Cheon, S. I., Silva, L. B. C., Khair, A. S. & Zarzar, L. D. Interfacially-adsorbed particles enhance the self-propulsion of oil droplets in aqueous surfactant. Soft Matter 17, 6742-6750 (2021).
 10. Jee, A. -Y., Tlustya, T. & Granick, S. Master curve of boosted diffusion for 10catalytic enzymes. Proc. Natl. Acad. Sci. U.S.A. 117, 29435-29441 (2020).
 11. MacDonald, T. S. C., Price, W. S., Astumian, R. D. & Beves, J. E. Enhanced Diffusion of Molecular Catalysts is Due to Convection. Angew. Chem. Int. Ed. 58, 18864–18867 (2019).

I have the feeling that there is something interesting in this paper, but the obscure language, jargon, and

overall unclear writing makes it almost impossible to uncover. This paper could possibly be rewritten with a view to clarity that would allow readers to see what is important, but that would require, in my opinion, a very major revision.

We thank the reviewer again for stressing this issue. We have significantly rewritten and reorganized the revised manuscript. We hope that this rewriting has made the manuscript clearer and better highlights the novel achievements of this work.

Reviewer #3 (Remarks to the Author):

The paper describes interested experimental observations that seem to be in line with our current understanding of diffusiophoresis. The authors explore a system where ATP conversion to ADP and AMP results in simultaneous strong effects regarding phoretic motion. The strong effects can be understood by the significant different mobilities that each compound presents for the particle phoresis.

The subtle and biological relevant dynamics that can be found in this system are interesting and worth publication. However, I feel that the authors could improve the explanation, especially quantitative, regarding the observed phenomena. For instance, drift distances are provided, as opposed to determining the phoretic mobilities. Theories are available, based on particle zeta potential, to predict the mobility based on chemo and electrophoretic contributions. These are provided in the SI, but no comparison has been made based on the observed drift distances/velocities in the other experiments (figure 3 and 4). Perhaps it is the difficulty in predicting/quantifying the concentration gradient during the experiments.

- We thank the reviewer for finding the manuscript interesting and worthy of publication.
- It is indeed difficult to directly model the experimentally observed values with theory, particularly as the system becomes more complex with multiple salts, and enzymatic reactions. However, in the revised manuscript, we have aimed to provide dimensional and scaling arguments to make a better connection between theory and experiments. For instance, we now add the following paragraph surrounding the discussion of Fig. 3:

“To understand the difference between AMP and ATP, we assume that the transport is controlled by the diffusion of nucleotides towards the left arm, which establishes gradients driving

diffusiophoresis of CMB. We thus expect that the distance traveled by the CMB after time t scales as $(2\Gamma_p t)^{1/2}$, where Γ_p is the diffusiophoretic mobility, and is typically a fraction of the ambipolar diffusivity D_a of the nucleotide (see SI for definitions). For AMP, $\Gamma_p \approx 5.7 \times 10^{-10} \text{ m}^2/\text{s}$, about ten times greater than in ATP (Supplementary Table 2). The estimated drift distance $(2\Gamma_p t)^{1/2}$ at the end of 5 minutes is on the order of one millimeter, consistent with experimental measurements. More importantly, the difference between AMP and ATP from this estimate is a little over 0.4 mm, which agrees well with the experiments (Fig. 3b). We additionally solve a one-dimensional numerical model of diffusiophoresis driven by the diffusion of nucleotides. Although the model is complicated by uncertainties associated with initial conditions (due to variability in the addition of the CMB), we find that it predicts that the CMB drift 0.3 – 0.4 mm more in the presence of AMP than with ATP, corroborating the scaling estimates, and in agreement with experiments (Fig. 3b, Supplementary Fig. 19, SI). This confirms that the diffusiophoresis, mediated by multivalency, is responsible for the differential transport of CMB in AMP versus ATP gradients in this non-continuous flow system.”

For ionic solutes, the phoretic velocity scales with the relative concentration gradient. This seems crucial for the time evolution of this velocity in a spatially changing concentration gradient.

- The referee is correct that the diffusiophoretic velocity scales with the relative (or logarithmic) concentration gradient, $\nabla c/c = \nabla (\ln c)$. This is accounted for throughout, and the details are described in Methods and Supplementary Information.

The microfluidic designs seem to pose some experimental challenges that the authors do not report on. For the drift experiments (figure 2) it will be very difficult to maintain the interface between the two solutions at exact position. Slight differences in flow rate affect the middle position and with that the determination of the drift distance. For the other experiments (figure 3 and 4), the establishment of the concentration gradient is not straightforward. Filling of the channel from one side, followed by filling the other reservoir for sure introduces some advective flow within the device. How has this been minimized, or quantified? The initial situation is crucial, as the concentration gradients at that moment determine the phoresis, but also set the spatial concentration gradients that follow.

- We thank the reviewer for raising this question. In each case for Figure 2 (continuous flow experiments), we made sure to maintain the interface between two solutions exactly at the centerline, up to a tolerance of at most 3 μm . The particle drift is much greater than this

tolerance, and is very consistent across experimental trials, as we show in our revised Supplementary Material. In the “Methods” section of the revised manuscript, we include the following description of our experimental protocol:

“During each measurement, we made sure the meeting point of the two flows at the inlet remain exactly at the center of the channel (Supplementary Figure 20, SI). Notably, the deviation at the inlet with respect to center of the channel and interface of two solutions is even less than $3\ \mu\text{m}$, which is much lower than our observed drift near the outlet (1.6 mm down-stream apart) in case of nucleotides.”

In the revised SI, we added the following figure.

Supplementary Fig. 20. Representative fluorescence image of the channel at the inlet showing that the drift of the fluorescent CMB showing the interface between two solutions with fluorescent CMB (on the bottom) and only a) water, b) AMP, c) ADP, d) ATP on the top of the channel. Please see Fig. 2a of the main manuscript for the experimental set up. Experimental set up details: Channel width = 0.6 mm, channel height = 0.1 mm, Flow velocity = $300\ \mu\text{l/h}$. Concentration of nucleotide in each case was 1 mM. The deviation at the inlet with respect to center of the channel and interface of two solutions is less than $3\ \mu\text{m}$, which is much lower than our observed drift near the outlet (1.6 mm downstream apart) for each case.

- For the cases of Figure 3 and 4 (non-continuous flow experiments), we first fill the tube slowly by adding nucleotide solutions and then allowed to settle it for 10 sec. Then, we added the fluorescent CMB particle from the other arm (which we define as time = 0) and started to record the positioning after 10 sec of CMB addition. We agree that there is likely to be some advective flow during the addition of the CMB. We account for this by using a control with only water solution inside the chamber (no nucleotides). With nucleotides, the observed drift is greater

than with only water. Additionally, we report the average of 5 experiments to decrease the error. In the manuscript Figure 3b, we report the *additional* drift in the presence of nucleotides, in excess of that observed with only water. However, in the Supplementary Information, we report the raw data, which we also show below:

Supplementary Fig. 18. Phoretic drift of CMB in macroscopic chamber in presence of different nucleotides and water. Experimental condition: 0.025 mg/mL bead, $[GNR] = 37.5 \text{ pM}$, $[Nucleotide] = 1 \text{ mM}$ at $25 \text{ }^\circ\text{C}$. The error bar is the standard deviation of 5 experiments.

The authors have ignored any diffusio osmotic flow caused by the gradient along the channel wall. This flow can also affect the particle motion and is not based on their phoresis. At least, the magnitude of this flow should be established.

- The referee raises an interesting point. We believe that diffusio-osmotic flows only have a weak effect on our experimental measurement, since the systems under consideration are closed, so there is no net diffusion-osmotic flux in the system. We detail this below.
- In the continuous-flow experiments, the drift is transverse to the flow. We also expect diffusio-osmotic (DO) slip to therefore occur in the cross-stream direction. However, because the channel is closed on the sides, these must be recirculating flows (since there cannot be any net flux in the cross-stream direction), affecting no *net* transport of particles. We illustrate this in the sketch below, indicating the qualitative structure of the DO flow in a cross-section transverse to the externally applied flow, which is out of the page towards the reader in the x direction (see Fig. 2a in the main text for the coordinate system). While DO may indeed modify the spatial distribution of particles across the height of the channel, we do not pick this up in our measurements since the fluorescence intensity (which is measured from below) is likely to be an average across the channel depth. We acknowledge that DO flows may produce Taylor-dispersion-like effects, though we expect these to be weak. Finally, the excellent agreement between theory (which does not model DO) and experiments (Fig. 2) without fitting parameters suggest that DO effects do not have a strong influence on particle motion.

The situation is similar for the setups with non-continuous flows, although their effect may be greater due to the somewhat larger dimensions.

REVIEWERS' COMMENTS

Reviewer #1 (Remarks to the Author):

The revised paper is clear and brings together several different topics - biological kinetics, colloid transport, diffusiophoresis - to show how these features can be tuned and controlled. The use of a system of AMP, ADP, and ATP with different valances, which then influence colloidal transport in chemical gradients is documented clearly. The further chemistry whereby ATP, which tends to reduce the chemical transport by binding to the surface and lowering the zeta potential, is converted to AMP, which is subsequently more active, is an interesting example of in situ transport modifications being driven by enzymatic chemistry in a controlled way. The use then show space and time control. As the authors document in the references, several of these individual pieces had been demonstrated previously but here they are integrated to achieve new understanding and highlight the potential of these ideas for controlled transport. The combination of experiments and quantitative modeling gives the paper added impact. I recommend publication.

Note: I did have one question. The author note how the different molecules bind to the particle/nano-rod covered surface, which changes the zeta potential. I formed the impression then that when particles move in a gradient of AMP/ADP/ATP there could be surface reactions modifying (or regulating) the surface zeta potential. However, reading the SI these seems not to be the case. Perhaps the authors can add a few words clarifying this aspect of the chemistry and modeling.

Reviewer #2 (Remarks to the Author):

The authors have, for the most part, satisfied my concerns in the revision of their paper. I remain skeptical about the use of jump (I think rapid increase would be a better description) to characterize the observation that there is a time lag until the phoretic velocity increases due to formation of AMP from ATP catalyzed by asparase. Nevertheless I find the paper very interesting, the experiments to be well thought out and performed and recommend publication in the present form.

Reviewer #3 (Remarks to the Author):

the authors have addressed my comments. I support publication.

RESPONSE TO REVIEWERS' COMMENTS

Reviewer #1 (Remarks to the Author):

The revised paper is clear and brings together several different topics - biological kinetics, colloid transport, diffusiophoresis - to show how these features can be tuned and controlled. The use of a system of AMP, ADP, and ATP with different valences, which then influence colloidal transport in chemical gradients is documented clearly. The further chemistry whereby ATP, which tends to reduce the chemical transport by binding to the surface and lowering the zeta potential, is converted to AMP, which is subsequently more active, is an interesting example of in situ transport modifications being driven by enzymatic chemistry in a controlled way. The use then show space and time control. As the authors document in the references, several of these individual pieces had been demonstrated previously but here they are integrated to achieve new understanding and highlight the potential of these ideas for controlled transport. The combination of experiments and quantitative modeling gives the paper added impact. I recommend publication.

- We thank the reviewer for finding the work impactful and recommending for publication.

Note: I did have one question. The author note how the different molecules bind to the particle/nano-rod covered surface, which changes the zeta potential. I formed the impression then that when particles move in a gradient of AMP/ADP/ATP there could be surface reactions modifying (or regulating) the surface zeta potential. However, reading the SI these seems not to be the case. Perhaps the authors can add a few words clarifying this aspect of the chemistry and modeling.

- In the revised manuscript, we added the following line at page 8, with proper citation, mentioning the following, “*We note that while the zeta potential may, in principle, vary as the CMB moves through the nucleotide concentration field, the present model yields accurate results for the drift without the need to account for this variation.*⁵⁰”

Reviewer #2 (Remarks to the Author):

The authors have, for the most part, satisfied my concerns in the revision of their paper. I remain skeptical about the use of jump (I think rapid increase would be a better description) to characterize the observation that there is a time lag until the phoretic velocity increases due to formation of AMP from ATP catalyzed by asparase. Nevertheless I find the paper very interesting, the experiments to be well thought out and performed and recommend publication in the present form.

- We are thankful to the reviewer for recommending publication in the present form.

Regarding the use of word 'jump': We actually defined in the introduction that this rapid increase in phoretic velocity is termed as 'phoretic jump', Below is the line:

“In particular, we developed an autonomous system that exhibits a rapid increase of phoretic velocity (which we term a phoretic jump) through enzymatic activity,.....”

Reviewer #3 (Remarks to the Author):

the authors have addressed my comments. I support publication.

- We are thankful to the reviewer for supporting publication of this article after revision.